# Tumor Microenvironment-Responsive Magnetic Nanofluid for Enhanced Tumor MRI and Tumor multi-treatments

**DOI:** 10.3390/ph16020166

**Published:** 2023-01-23

**Authors:** Liangju Sheng, Xuanlei Zhu, Miao Sun, Zhe Lan, Yong Yang, Yuanrong Xin, Yuefeng Li

**Affiliations:** 1College of Medicine, Jiangsu University, Zhenjiang 212013, China; 2College of Pharmacy, Jiangsu University, Zhenjiang 212013, China; 3College of Mechanical Engineering, Suzhou University of Science and Technology, Suzhou 215009, China

**Keywords:** tumor microenvironment-responsive, magnetic nanofluid, improved magnetic resonance imaging, neutral-responsive Fenton reaction, enhanced chemodynamic therapy

## Abstract

We prepared a tumor microenvironment-responsive magnetic nanofluid (MNF) for improving tumor targeting, imaging and treatment simultaneously. For this purpose, we synthesized sulfonamide-based amphiphilic copolymers with a suitable p*K*_a_ at 7.0; then, we utilized them to prepare the tumor microenvironment-responsive MNF by self-assembly of the sulfonamide-based amphiphilic copolymers and hydrophobic monodispersed Fe_3_O_4_ nanoparticles at approximately 8 nm. After a series of characterizations, the MNF showed excellent application potential due to the fact of its high stability under physiological conditions and its hypersensitivity toward tumor stroma by forming aggregations within neutral or weak acidic environments. Due to the fact of its tumor microenvironment-responsiveness, the MNF showed great potential for accumulation in tumors, which could enhance MNF-mediated magnetic resonance imaging (MRI), magnetic hyperthermia (MH) and Fenton reaction (FR) in tumor. Moreover, in vitro cell experiment did not only show high biocompatibility of tumor microenvironment-responsive MNF in physiological environment, but also exhibit high efficacy on inhibiting cell proliferation by MH-dependent chemodynamic therapy (CDT), because CDT was triggered and promoted efficiently by MH with increasing strength of alternating magnetic field. Although the current research is limited to in vitro study, these positive results still suggest the great potential of the MNF on effective targeting, diagnosis, and therapy of tumor.

## 1. Introduction

In tumor diagnosis and treatment, superparamagnetic iron oxide (SPIO) nanoparticles play a unique and important role, because they possesses versatile applications for clinical diagnosis and tumor adjunctive therapy [1,2]. For tumor diagnosis, SPIO nanoparticles have been used widely as a contrast agent (CA) in magnetic resonance imaging (MRI), as it could improve the contrast in anatomical imaging to highlight the situation and structure of a tumor by shortening the spin−spin relaxation time (*T*_2_) of the proton [3]. For tumor therapy, magnetic hyperthermia (MH) is a noninvasive hyperthermia that inhibits tumor growth by the Brownian relaxation and Néel relaxation of SPIO nanoparticles under an alternating magnetic field (AMF) [4]. Furthermore, many studies also found that SPIO nanoparticles could induce apoptosis of tumor cells directly by producing ferrous ions, which can generate toxic reactive oxygen species (ROS) by the Fenton reaction (FR) [5]. Although SPIO nanoparticles present much potential, their effectiveness in tumor diagnosis and therapy still depends on their accumulation in the tumor, which is similar to that of a chemotherapeutic drug. Learning from the progress of stimuli-responsive polymeric nanocarriers for tumor targeting [6,7], the targeted accumulation of SPIO nanoparticles in a tumor could also be improved by constructing stimuli-responsive magnetic nanofluid (MNF) through the self-assembly of SPIO nanoparticles and stimuli-responsive polymers.

Because tumor angiogenesis and aerobic glycolysis have been recognized as features of most malignant solid tumors, regardless of their tissue origin or genetic background [8], pH-sensitive nanocarriers have attracted tremendous interest in tumor diagnosis and subsequent treatment over the past decades [9,10]. In order to construct pH-sensitive nanocarriers, many types of pH-sensitive polymers have been designed and synthesized, as they possess ionizable basic or acidic residues [11], resulting in varied physicochemical properties (solubility or chain conformation) with a change in the surrounding pH [12]. However, slight pH differences between a tumor stroma (pH = 7.1–6.7) [13] and the physiological condition (pH = 7.35–7.45) is impossible to be recognized by most pH-sensitive deliveries, which usually consist of pH-sensitive polymers and occur during phase transition in endocytic organelles with a lower pH value (pH ≤ 6.0), such as endosomes and lysosomes [9,10]. There are only a few pH-sensitive polymers with dissociation constants (p*K*_a_) around neutral pH, such as cationic polymers with repeating ionizable tertiary amines [14,15] or anionic polymers with suitable sulfonamide groups [16,17]. Until now, the most successful pH-sensitive polymers used in targeting tumor microenvironment were cationic polymers with ionizable tertiary amine blocks, showing a hydrophobic–hydrophilic phase transition in weakly acidic microenvironments [18,19], which disassemble rapidly in tumor microenvironment, leading to their application in targeted tumor chemotherapy [20] and enhanced tumor fluorescence imaging [19]. Unfortunately, ionized cationic polymer exhibit a potential risk of hemolysis in vivo, as its positive charge could easily damage membranes of red blood cells [21]. Therefore, a sulfonamide-based anionic polymer with suitable p*K*_a_ within a tumor microenvironment pH value (7.1–6.7) should be an optimized option to prepare MNF with tumor microenvironment responsiveness.

According to the property of an anionic polymer, its hydrophilic–hydrophobic phase transition can be triggered under a certain pH value, which is usually lower than the p*K*_a_ of anionic polymer [11]. As a result, a pH-responsive MNF with a suitable p*K*_a_ (7.1–6.7) should lose its colloidal stability in tumor stroma, resulting in the efficient accumulation and retention of SPIO in tumor tissue. Furthermore, tumor tissue had existing amounts of endogenous hydrogen ion (H^+^) and hydrogen peroxide (H_2_O_2_). Due to the phase transition of anionic polymers in tumor stroma, encapsulated SPIO nanoparticles obtained the opportunity to interact with surrounding H^+^, resulting in the release of ferrous ion (Fe^2+^). Next, the endogenous H_2_O_2_ in the tumor tissue could be decomposed under the catalysis of Fe^2+^, implying the possibility of FR in the tumor microenvironment. When the phase transition of anionic polymer blocks occurred completely, stranded SPIO nanoparticles could form many aggregations with a large size spontaneously. According to previous studies, a closed packing structure of the multiple SPIO nanoparticles exhibited its attractive effects on improving the negative signal contrast of pathological tissue [22] and enhanced the efficiency of MH [23] simultaneously. Therefore, sulfonamide-based MNF with a pH responsiveness not only possesses a specific advantage in tumor targeting but also shows other potentials for improving the sensitivity of tumor MRIs and enhancing the antitumor efficacy by a combination of MH- and FR-mediated chemodynamic therapy (CDT).

In this study, we designed and fabricated a tumor microenvironment-responsive MNF, which maintained stability in blood vessels and formed aggregations in tumor microenvironment to enhance tumor MH, ROS generation and MRIs simultaneously, as shown in Figure 1.

In order to prepare the tumor microenvironment-responsive MNF, we synthesized a sulfonamide-based amphiphilic copolymer, polycaprolactone-*b*-poly(sulfadimethoxine acrylamide) (PCL-*b*-pSMA) by reversible addition−fragmentation chain transfer (RAFT) polymerization, according to relevant studies [24,25], which showed p*K*_a_ of approximately 7.0. Then, the MNF was fabricated by the simple self-assembly of the PCL-*b*-pSMA and hydrophobic Fe_3_O_4_ nanoparticles. Due to the fact of the pH sensitivity of PCL-*b*-pSMA at a neutral pH value, the MNF in the aqueous phase displayed a similar pH responsiveness at a neutral pH value (≈7.0). According to the pH sensitivity of the MNF on neutral medium, we further investigated its application potentials for MH, FR and MRI for tumor theranostics. The relevant results show that the MNF under neutral conditions exhibited a better performance in enhancing the specific absorption rate (SAR), increasing the generation of hydroxyl radical (•OH) and improving the *T*_2_ relaxivity (*r*_2_), simultaneously, compared to its counterpart in a physiological environment. Based on these advantages, we further studied the effects of the tumor microenvironment-responsive MNF on MH-induced cell death and ROS generation under different strengths of AMF (*H*_applied_). As the intercellular ROS level and cell mortality rate under MH showed a high degree of correlation, this study suggests that the MNF can stimulate CDT by MH and inhibit cell proliferation efficiently by integrating MH and CDT effectively. Therefore, the tumor microenvironment-responsive MNF possesses a versatile potential for tumor targeting, diagnosis and treatment.

## 2. Results and Discussion

### 2.1. Synthesis and Characterization of the pH-Responsive Amphiphilic Copolymer

As an anionic polymer usually possesses hydrophilicity in neutral and physiological conditions, we selected the polycaprolactone (PCL) segment as the hydrophobic block of the amphiphilic copolymer because of its high biocompatibility. The synthetic route, as shown in Appendix A, contained ring opening polymerization (ROP), an esterification reaction and RAFT polymerization simultaneously. By the ROP, the PCL was synthesized successfully, which was confirmed by the H proton nuclear magnetic resonance (^1^H NMR) spectrum, as shown in Appendix A. S-1-dodecyl-S’-(a,a’-dimethyl-a”-acetic acid)trithiocarbonate (DDMAT) was used as the chain transfer agent (CTA) for the RAFT polymerization. The following product was PCL-DDMAT by the esterification reaction between PCL and DDMAT, which was also confirmed by its structure using the ^1^H NMR spectrum, as shown in Appendix A.

The final reaction was the RAFT polymerization for the preparation of the pH-responsive amphiphilic copolymer. In this study, we selected sulfadimethoxine acrylamide (SMA) as the pH-sensitive monomer, because the poly(methacryloyl sulfadimethoxine) showed a p*K*_a_ at approximately 7.0 in a relevant study [24]. Utilizing DMSO-*d*_6_ as a solvent, we characterized the structures of the PCL-*b*-pSMA and SMA, as shown in Figure 2. According to previous studies [24,26,27], all of the characteristic peaks of the SMA were identified and marked in Figure 2A (bottom spectrum). Based on the ^1^H NMR results of the SMA (Figure 2A, bottom spectrum) and PCL-DDMAT (Appendix A), we further identified all of the characteristic peaks of the PCL-*b*-pSMA, also shown in Figure 2A (top spectrum). Apparently, because of the polymerization of SMA, the characteristic peaks of the acrylamide in the SMA disappeared; meanwhile, all of the characteristic peaks of the SM broadened. In addition to the characteristic peaks of SM, the characteristic peaks of PCL can also observed in Figure 2A (top spectrum), which confirms the successful polymerization of SMA as a product of the PCL-DDMAT.

After researching the structure of the PCL-*b*-pSMA, we characterized the p*K*_a_ of the PCL-*b*-pSMA, as shown in Figure 2B. According to relevant studies [28,29], the cloud point method was utilized to determine the p*K*_a_ of the PCL-*b*-pSMA, as shown in Figure 2B (red line), by quantifying the turbidity of the PCL-*b*-pSMA under different pH buffers at 500 nm. We prepared the micelles of the PCL-*b*-pSMA in an alkaline solution first; then, we observed its light transmittance using a UV-Vis spectrophotometer under a decreasing pH from 9.13 to 5.18. Apparently, the transmittance of the PCL-*b*-pSMA micelles with a high concentration (≈3 mg mL^−1^) was influenced by the environmental pH value. In the alkaline environment, the PCL-*b*-pSMA micelles showed almost 100% transmittance. When the pH value decreased to 7.5, its transmittance decreased slightly. However, the turbidity increased sharply with the decrease of the pH from 7.27 to 6.88, and the corresponding light transmittance decreased from 87.7% to 29.3%. When the pH value decreased further (≤6.59), the PCL-*b*-pSMA formed obvious sediment, and the corresponding light transmittance was near 0%. As PCL is a typical hydrophobic polymer, the high transmittance of the PCL-*b*-pSMA depends on the hydrophilicity of the pSMA in an alkaline solution. When the solution pH value downregulated from a weak alkaline to faintly acid, the increasing turbidity of the PCL-*b*-pSMA micelles indicated the rapid phase transition of the pSMA from hydrophilicity to hydrophobicity.

The phase transition of the pSMA not only decreased the transmittance of the PCL-*b*-pSMA micelles at the macro level but also reduced their stability at the micro level, which could be observed by dynamic light scattering (DLS), which was assessed under a low concentration of the PCL-*b*-pSMA micelles (0.3 mg mL^−1^). As shown in Figure 2B (gray line), the DLS result exhibited the effect of the pH value on the PCL-*b*-pSMA micelle’s hydrated diameter. According to the result, the hydrated diameter of the PCL-*b*-pSMA micelles in an alkaline solution decreased slightly from 64.7 (pH ≈ 9.21) to 62.9 nm (pH ≈ 7.43) with the decreasing pH value. When the surrounding pH value decreased to 7.05, the diameter of the PCL-*b*-pSMA micelles increased sharply to 198.9 nm. With the pH decreasing further (pH ≈ 6.85), the particle size of the PCL-*b*-pSMA micelles increased to 379.4 nm, indicating the agglomeration of the PCL-*b*-pSMA micelles. Finally, the diameter of these agglomerated micelles reached almost 600 nm in an acidic environment, which corresponded to the very low transmittance of its counterpart, with 10 times the concentration. According to the results shown in Figure 2B, we estimated the p*K*_a_ value of the PCL-*b*-pSMA at 7.0, because the PCL-*b*-pSMA micelles presented 50% transmittance at that pH value.

Furthermore, we observed the PCL-*b*-pSMA micelles under a physiological and neutral pH value directly by transmission electron microscopy (TEM), as shown in Figure 3A,B. By the negative staining of phosphotungstic acid, many bright spheres could easily be observed, which represented the inner core of the PCL-*b*-pSMA micelles. Apparently, the PCL-*b*-pSMA micelles in the buffer with a physiological pH value displayed high colloidal stability, because their inner cores (bright spheres) could be separated from each other by a hydrophilic shell composed of pSMA. On the contrary, in the neutral buffer (pH ≈ 7.05), the PCL-*b*-pSMA micelles did not display a larger inner core, but also collected together spontaneously, as shown in Figure 3B, indicating that the colloidal stability of the PCL-*b*-pSMA micelles was broken.

Based on the TEM results, we investigated these TEM samples again by digital photos and DLS again; all are shown in Figure 3C. It was clear that the PCL-*b*-pSMA micelles in neutral solution looked similar to a milk solution; meanwhile, its counterpart in the physiological buffer with the same concentration showed excellent transmittance. Corresponding to these photos, the PCL-*b*-pSMA micelles in the physiological buffer in the neutral solution presented a smaller diameter (62.9 nm) and a narrower particle size distribution (PDI = 0.141) than their counterparts in the neutral solution, which explains the high transmittance of the former and the low transmittance of the latter.

Due to the fact of the results of the transmittance, DLS, TEM and digital photos, we prepared an anionic pH-responsive amphiphilic copolymer with a p*K*_a_ of approximately 7.0, which should be suitable for the preparation of a tumor microenvironment-responsive MNF.

### 2.2. Characterization of the Tumor Microenvironment -Responsive MNF

In order to prepare the tumor microenvironment-responsive MNF, we firstly synthesized hydrophobic Fe_3_O_4_ nanoparticles at approximately 8 nm. After, the MNF was prepared by the simple self-assembly between the amphiphilic PCL-*b*-pSMA and hydrophobic Fe_3_O_4_ nanoparticles. It was clear that the Fe_3_O_4_ nanoparticles in hexane (0.1 mg mL^−1^) presented monodispersity with a uniform particle size, as shown in Figure 4A, due to the fact of their high hydrophobicity. In order to improve the water dispersibility of the hydrophobic Fe_3_O_4_ nanoparticles, the polymeric micelles provided a powerful platform to load the hydrophobic molecules into their hydrophobic cores. Therefore, we prepared the MNF by the self-assembly of the hydrophobic Fe_3_O_4_ nanoparticles and the amphiphilic PCL-*b*-pSMA, with a corresponding weight ratio of 3/7. The obtained pH-responsive MNF could be dispersed in an alkaline buffer (pH ≈ 9.0) directly. We further observed the morphology of the MNF at a certain concentration (0.5 mg mL^−1^), which is shown in Figure 4B. Apparently, in the aqueous phase, the hydrophobic Fe_3_O_4_ nanoparticles formed round packed clusters unlike the monodispersed nanoparticles in hexane (Figure 4A), which confirmed the successful fabrication of micellar MNF by encapsulation of the PCL-*b*-pSMA.

In addition to TEM, we also used DLS to characterize the diameters of the Fe_3_O_4_ nanoparticles and MNF, as shown in Figure 4C. Apparently, the DLS result confirmed the Fe_3_O_4_ nanoparticles with a uniform particle size of approximately 8 nm, again, because of their low particle size distribution (PDI = 0.125). In addition, the DLS result also confirmed that the MNF possessed a larger particle size (145.2 nm) and a wider particle size distribution (PDI = 0.145) than those of the Fe_3_O_4_ nanoparticles, which corresponded to the TEM results in Figure 4A,B. Furthermore, the magnetic hysteresis curves of the Fe_3_O_4_ nanoparticles and MNF were characterized, as shown in Figure 4D. The Fe_3_O_4_ nanoparticles presented obvious superparamagnetism, corresponding to their diameters at approximately 8 nm. Consequently, the MNF also showed superparamagnetism without significant remanent magnetization. Meanwhile, the Fe_3_O_4_ nanoparticles exhibited a high saturation magnetism (*M*_s_ = 60.4 emu g^−1^); on the contrary, the MNF showed a relatively low high saturation magnetism (*M*_s_ = 17.9 emu g^−1^). To understand the phenomenon, we measured the Fe_3_O_4_ content in the MNF by thermogravimetric analysis (TGA). As shown in Appendix A, the content of the Fe_3_O_4_ nanoparticles was 29.3 wt%, which was close to its feed ration. Therefore, it was reasonable that the saturation magnetism of the MNF was 30% of that of the Fe_3_O_4_ nanoparticles. According to these results, the MNF was prepared successfully by loading the hydrophobic Fe_3_O_4_ nanoparticles into the inner core of the polymeric micelles.

In order to verify the pH responsiveness of the MNF, we also studied the macroscopic states and microscopic morphologies of the MNF in different buffers with the same concentration (0.5 mg mL^−1^), as shown in Figure 5.

Corresponding to a p*K*_a_ value of the PCL-*b*-pSMA, the MNF could disperse very well in the aqueous phase with a physiological pH value (pH ≈ 7.43), forming a transparent brown liquid in the macroscopic state. Furthermore, we assessed the long-term stability of the MNF (1~13 d) under a physiological pH value (≈7.43), as shown in Appendix A, which confirmed its high colloidal stability under physiological conditions. At the same time, the MNF solution with a neutral pH value (pH ≈ 7.02) became a dark-brown liquid at the macroscopic level. With the further decrease in the pH value, the MNF lost its colloidal stability in the solution with weak acidity (pH ≈ 6.77), as shown in Figure 5A; many sediments could be observed after adjusting the pH to 6.77 for 2 min. With the prolonged observation time of 30 min, in physiological and neutral conditions, the MNF maintained its original states; however, the MNF in a weak acidic environment dropped completely to the bottom of the container. In addition to the differences at the macroscopic level, the MNF solutions with corresponding pH values for 2 min also showed quite different microscopic morphologies, as shown in Figure 5B. The MNF in the aqueous phase with pH = 7.43 exhibited a similar morphology as the TEM result in Figure 4B, indicating its colloidal stability under the physiological conditions again. However, in a neutral environment, several micellar MNF formed a large Fe_3_O_4_ cluster, as shown in Figure 5B, resulting in the low transmittance of the sample. When the surrounding pH value was downregulated to weak acidity (pH ≈ 6.77), many Fe_3_O_4_ nanoparticles piled up chaotically, also shown in Figure 5B, because the surrounding pH value was lower than the p*K*_a_ of the PCL-*b*-pSMA. In addition to the macroscopic states and microscopic morphologies, we also quantified the diameters (Appendix A) and zeta potentials (Appendix A) of the MNF in solutions with corresponding pH values, which are shown in Appendix A. Apparently, in a neutral and weak acid solution, the MNF not only displayed larger diameters but also showed higher surface potentials (negative charge) compared to their counterparts in a physiological environment. According to these results, the MNF exhibited a similar pH responsiveness with a neutral pH value, which can be ascribed to the p*K*_a_ of the PCL-*b*-pSMA. Considering a tumor microenvironment pH value (7.1~6.7) [13], the tumor microenvironment-responsive MNF should possess a potential for targeted accumulation in a tumor stroma.

### 2.3. Advantages of Tumor Microenvironment-responsive MNF for Tumor Treatment and Diagnosis

Due to the high magnetic particle content, we studied the magnetocaloric effects of the MNF under different conditions, and the corresponding heating curves are shown in Figure 6A.

Under the same concentration of Fe (100 μg mL^−1^), the MNF could increase the surrounding temperature rapidly in solutions with different pH values; however, the fastest heating rate and highest heating temperature occurred in neutral conditions simultaneously. Additionally, the initial heating rate and final heating temperature of the MNF in a weak acid environment were also higher than its counterpart in physiological conditions. Therefore, the ranking of the MNF’s SAR in the different buffers is neutral condition > weak acid > physiological condition, which is also shown in Figure 6A. This is an interesting phenomenon, because environmental factors of MH are rarely reported. According to the mechanism of MH, the monodispersed SPIO nanoparticles with a small diameter (≤10 nm), as used in this study, induced a magnetocaloric effect under AMF mostly by Néel relaxation, which is associated with magnetic moment [4]. Therefore, many previous studies have focused on material factors for enhancement of the magnetic moment of SPIO, including composition [30], shape [31], and structure [32]. However, in the past decade, many relevant studies also found that the micellar MNF with a large diameter exhibited high efficiency in producing thermal energy by Néel relaxation [23,33], because the effective magnetic moment of the SPIO could be improved by increasing the diameter of the magnetic nanocluster. In this study, due to the results shown in Figure 5B and Appendix A, the tumor microenvironment-responsive MNF in neutral and weak acid conditions displayed a larger diameter by the aggregation of multi-magnetic micelles, compared to its counterpart in a physiological environment. Consequently, the magnetocaloric effect of the MNF could be enhanced specifically in tumor stroma, which should benefit the application of tumor microenvironment-responsive MNF for clinical MH. However, it is worth noting that the tumor microenvironment-responsive MNF showed the highest SAR value under neutral conditions. This should be attributed to the Brownian relaxation of the MNF under AMF, because the agglomerated MNF could maintain dispersibility in the neutral solution, as shown in Figure 5, leading to their free rotation under AMF. Therefore, the tumor microenvironment-responsive MNFs could generate thermal energy efficiently in the neutral solution by integrating Brownian relaxation and Néel relaxation. On the contrary, in a weak acid environment, the MNF could not undergo Brownian motion easily, because it formed precipitates rapidly. Consequently, the MH of the MNF in the weak acid solution could be induced by Néel relaxation alone.

In addition to MH, the neutral condition could also improve the catalytic performance of the MNF to generate •OH, as shown in Figure 6B. It is well known that Fe^2+^ can decompose low toxic H_2_O_2_ to form high toxic •OH by FR efficiently [34,35], inducing the application of the SPIO nanoparticles for CDT in many previous studies [36]. Therefore, in this study, we evaluated the •OH-generating ability of the MNF in solutions with different pH values using methylene blue (MB) as a detection probe, since the MB could be decomposed by •OH. Obviously, H_2_O_2_ could not break the structure of the MB, as the UV spectrum of the MB alone (a) coincided with that of the mixture of MB and H_2_O_2_ (b). However, by incubating with H_2_O_2_ and the MNF (c~e) simultaneously, the UV absorbance of the MB declined, indicating its degradation. Moreover, the surrounding pH value influenced the degradation of the MB significantly, also shown in Figure 6B, in which a neutral environment (d) was the best condition for the catalytic performance of the tumor microenvironment-responsive MNF. This was an unexpected result, because most previous studies showed that an acidic environment (pH ≤ 6.0) could improve the efficiency of FR [34,37]. However, in these studies, the corresponding iron-based nanocatalysts maintained a high colloidal stability in a neutral environment, which reduced the interaction between the naked SPIO nanoparticles and H_2_O_2_. In our study, due to the phase transition of the PCL-*b*-pSMA, the interactions between H_2_O_2_ and the naked SPIO increased. More importantly, as shown in Figure 5, the agglomerated MNF could maintain the dispersed state in the aqueous phase for long time in a neutral environment; meanwhile, its counterpart under weak acidic conditions precipitated rapidly. This phenomenon indicated that the tumor microenvironment-responsive MNF presented a state of transition in a neutral environment, which increased the probability of an interfacial reaction between the Fe_3_O_4_ nanoparticles and H_2_O_2_ dramatically. Considering a neutral or weak acidic microenvironment of a tumor stroma, the tumor microenvironment-responsive MNF possesses a unique advantage for tumor microenvironment-responsive CDT.

The agglomerated MNF in a neutral environment not only presented an excellent catalytic performance but also enhanced the *T*_2_ relaxivity in the MRI, which is shown in Figure 6C,D. For a given Fe concentration, the *T*_2_ imaging of the MNF under neutral conditions (pH ≈ 6.99) was significantly darker than its counterpart in a physiological environment, as shown in Figure 6C. As the *T*_2_ imaging of the MRI represented a negative signal contrast, the darker imaging MNF in neutral conditions indicated the enhancement of the *T*_2_ imaging. Based on the results in Figure 6C, we quantified the *T*_2_ relaxivities of the MNF under different conditions, as shown in Figure 6D. The MNF in neutral conditions possessed a steeper slope (*r*_2_ = 271.6 mM^−1^s^−1^) compared to its counterpart in physiological conditions (*r*_2_ = 164.1 mM^−1^s^−1^). The results of the *T*_2_ imaging enhancement correspond to many relevant studies [22,38,39,40]; all of them confirmed that the *T*_2_ relaxation rate of the MRI could be enhanced by forming SPIO clusters and increasing the diameter of these clusters. Although the MNF was a micellar cluster of SPIO nanoparticles in a physiological environment, as shown in Figure 5, the particle size of the SPIO clusters could be increased further in neutral conditions. Therefore, the *T*_2_ imaging of the MNF could be enhanced by a neutral environment, indicating its application for tumor detection by MRI.

### 2.4. MH-Induced Apoptosis and Intracellular •OH Generation In Vitro

As it is hard to distinguish the pH value of a physiological condition (pH ≈ 7.35–7.45) and a neutral condition (pH ≈ 7.0) by RPMI-1640 medium, the effect of the pH value was not considered in the cell experiment.

The Cytotoxicity experiments were studied using 4T1. Firstly, we assessed the biocompatibilities of the pH-responsive micelles and MNF, as shown in Figure 7A. It was clear that under a physiological condition (pH ≈ 7.35–7.45), the pH-responsive micelles displayed biocompatibility, as its cell survival rates at all concentrations (0.025–1 mg mL^−1^) ranged from 100% to 90%. Compared to the pH-responsive micelles, the pH-responsive MNF showed a higher cell survival rate under same the concentration, also shown in Figure 7A, indicating the excellent biocompatibility of the MNF under physiological conditions.

Apparently, the MNF showed excellent biocompatibility; however, the MNF-based MH could also inhibit cell proliferation efficiently, as shown in Figure 7B. For these studies, the concentration of the MNF was fixed at 0.2 mg mL^−1^. After incubation with the MNF alone for 12 and 24 h, the 4T1 still displayed a high cell viability, which was similar as the result in Figure 7A. Nevertheless, after exposure to AMF with different *H*_applied_ for 10 min, the cell viabilities decreased obviously. Moreover, the inhibition rate of the 4T1’s proliferation under AMF exhibited a *H*_applied_-dependent tendency. When the *H*_applied_ of the AMF was fixed at 21.2 kA m^−1^, 4T1’s viabilities decreased to 81.2% at 12 h and 76% at 24 h after MH for 10 min. When the *H*_applied_ of the AMF was increased to 31.8 kA m^−1^, 4T1’s viabilities were suppressed exponentially, which was 21.4% for 12 h and 2.8% for 24 h. Therefore, the survival of the 4T1 under the AMF with the highest *H*_applied_ (42.4 kA m^−1^) decreased to 1.7% at 24 h after MH. To determine the relationship between the *H*_applied_ of the MH and the cell survival rate, we recorded the heating curves of the MNF under AMF with the same condition as described in the cell experiment, and the results are shown in Appendix A. Apparently, under the AMF with the given *H*_applied_ in this study, MH could not induce a sufficient temperature (>45 °C) to suppress cell death efficiently. Therefore, we evaluated the intercellular ROS level, another possible mechanism related to cell death, after different treatments.

As 2’,7’-dichlorodihydrofluorescin diacetate (DCFH-DA) can be metabolized within the cell by intercellular ROS, forming a fluorescent compound, 2’,7′-dichlorofluorescein (DCF), flow cytometry was utilized to quantify the intercellular ROS level by detecting the fluorescent intensity of the DCF. In order to correspond to the cytotoxicity of the MNF for different culture times, we studied the intercellular ROS level after incubating with MNFs for 12, 24 and 48 h. The result confirmed the excellent biocompatibility of the MNF, again, as shown in Figure 7C. The fluorescent intensities of the DCF for all predetermined times overlapped with that of 4T1 under standard culture conditions for 48 h. Although the MNF showed a high biocompatibility, MNF-mediated MH could boost the intercellular ROS level, as shown in Figure 7D. Further, the intercellular ROS level under MH presented a similar trend as that of its counterpart for the cell inhibition rate. For the group of MH-1, the low *H*_applied_ (21.2 kA m^−1^) limited the increase in the DCF fluorescent intensity, resulting in a high cell viability, as shown in the biocompatibility for the MNF. For the groups of MH-2 (31.8 kA m^−1^) and MH-3 (42.2 kA m^−1^), they showed similar intercellular ROS levels, corresponding to their cell mortality rate efficiencies on inhibiting 4T1 proliferation. Therefore, the SPIO biocompatibility did not conflict with the SPIO-mediated CDT. The occurrence of CDT should be triggered by a certain stimulation, such as MH [41], photothermal treatment [42], photodynamic therapy [43], and tumor microenvironment [44]. On the basis of the pH responsiveness under neutral conditions, the MNF possessed many advantages for tumor microenvironment-enhanced MH, catalytic activity and MRI. This novel MNF should be a competitive candidate for tumor diagnosis and treatment.

## 3. Materials and Methods

### 3.1. Materials

The Sn(Oct)_2_ (92.5–100%), 4,4’-Azobis(4-cyanovaleric acid) (V501, 98%), 1,2-hexadecanediol (97%) and Oleylamine (>70%) were purchased from Sigma Aldrich (Steinheim, Germany). ε-Caprolactone (ε-CL, 99%), N,N’-dicyclohexylcarbodiimide (DCC, 98%), 4-dimethylaminopyridine (DMAP, 99%) and sulfadimethoxine (SM, 98%) were purchased from Tokyo Chemical Industry (TCI, Tokyo, Japan). Iron(III) acetylacetonate [Fe(acac)_3_], benzyl ether (99%) and oleic acid (90%) were purchased from Alfa-Aesar (Heysham, England). Benzyl alcohol (BaOH, 99%, safe dry), acryloyl chloride (98%), tetrahydrofuran (THF, 99%) and MB (95%) were purchased from Admas (Shanghai, China). The dialysis tubing (8000–14,000 Da), dichloromethane (DCM, 99.9%), dioxane (99%), ethyl ether (99.5%), methanol (99.5%) and H_2_O_2_ (30%) were purchased from Sinopharm Chemical Reagent Co., Ltd. (Shanghai, China). 3-(4,5-Dimethyl-thiazol-2-yl)-2,5-diphenyl tetrazoliumbromide (MTT) was purchased from Beyotime Biotech. Co., Ltd. (Shanghai, China).

The SMA was synthesized by the procedure described in the relevant literature [45]. The DDMAT was synthesized according to [27], and the monodisperse superparamagnetic Fe_3_O_4_ nanoparticles were synthesized according to [46], with minor modifications. The typical synthetic procedure is described as follows: A certain amount of Fe(acac)_3_, 1,2-hexadecanediol, oleic acid and oleylamine with molar numbers of 2, 10, 2 and 2 mmol were dispersed successively in benzyl ether (20 mL). After deoxidizing by argon at 50 °C for 30 min, the mixture was heated to 200 °C for 2 h under an argon atmosphere and then heated further to reflux (≈300 °C) for two and a half hours. The product, monodispersed Fe_3_O_4_ nanoparticles, was precipitated by excess ethanol and then collected by centrifugation. The purified process was repeated three times. The purified Fe_3_O_4_ nanoparticles were dried by a high-purity argon flow. Finally, the magnetic nanoparticles were dispersed in THF with a concentration of 10 mg mL^−1^ for storage under −20 °C.

The other reagents were used as received. The water used in all experiments was deionized with a Millipore Milli-Q system (Billerica, USA).

### 3.2. Synthesis of the pH-Responsive Amphiphilic Copolymer

The pH-responsive amphiphilic copolymer, PCL-*b*-pSMA, was synthesized by the reaction procedure, as shown in Appendix A, which was described as follows.

In the first step, the PCL was synthesized by ROP using BaOH as an initiator and ε-CL as monomers with a corresponding molar ratio of 1:50. The reaction was heated to 110 °C under an argon (Ar_2_) atmosphere for 24 h. The product, PCL, was purified by precipitating in excess ethyl ether three times from its DCM solution. Finally, the PCL was dried until reaching a constant weight in a vacuum oven at an ambient temperature.

The second reaction was to prepare the macro-CTA, PCL-DDMAT, by esterification between the PCL and DDMAT, according to the established method [25]. In the reaction, a certain amount of PCL, DDMAT, DCC and DMAP with a corresponding molar ratio of 1:5:5:1 was weighted accurately and dissolved in anhydrous DCM by magnetic stirring. The esterification was carried out under an Ar_2_ atmosphere for 48 h. To purify the PCL-DDMAT, the supernatant of the reaction was collected and successively precipitated in excess cold ethyl ether. The process for the purification was repeated several times until the supernatant was without any DDMAT. Finally, the purified PCL-DDMAT was dried in a vacuum oven at an ambient temperature and preserved in Ar_2_ under a low temperature (−20 °C).

The third reaction was the synthesis of the PCL-*b*-pSMA by a RAFT reaction. In the reaction, the PCL-DDMAT (0.02 mmol, 110 mg), SMA (2 mmol, 728 mg) and V501 (0.004 mmol, 1.1 mg) were dissolved in dioxane of 5 mL under magnetic stirring. Then, the mixture was thoroughly degassed by three freeze–pump–thaw cycles. The reaction was carried out at 70 °C for 10 h. The final product, PCL-*b*-pSMA, was purified by repeated precipitation in excess methanol, followed by freeze-dried treatment and also preserved in Ar_2_ under a low temperature (−20 °C).

### 3.3. Characterization of the pH-Responsive Amphiphilic Copolymer

All products were characterized by their ^1^H NMR spectrum, with CDCl_3_ or DMSO-*d_6_* as the solvent, and their chemical shifts relative to tetramethylsilane (TMS) were identified. In order to identify the pH responsiveness of the PCL-*b*-pSMA, we prepared micelles of the PCL-*b*-pSMA first, where the THF solution of the PCL-*b*-pSMA (20 mg mL^−1^) was dialyzed against an alkaline solution (pH = 9.13) for 48 h. After, the obtained micelle solution of the PCL-*b*-pSMA with a high concentration (>3 mg mL^−1^) downregulated its pH value by the gradual addition of a small amount of hydrochloric acid solution (1 M). By using a UV-Vis spectrophotometer (Shimadzu, UV2600, Tokyo, Japan), we recorded its light transmittance at different pH values (9.13~5.18). The p*K*_a_ value of the PCL-*b*-pSMA was defined as the surrounding pH value, producing a 50% decrease in the optical transmittance at 500 nm [29]. At the same time, the p*K*_a_ value of the PCL-*b*-pSMA was also identified by DLS (Malvern, Nano ZS90, Worcestershire, UK), which measured the size distribution of the dilute PCL-*b*-pSMA micelles (0.3 mg mL^−1^) with varied surrounding pH values. Moreover, the morphologies of the PCL-*b*-pSMA micelles at pH values under physiological and neutral conditions were observed directly by TEM (Hitachi, HT-7800, Tokyo, Japan), in which the samples were stained by phosphotungstic acid (2%).

### 3.4. Preparation of the Tumor Microenvironment-responsive MNF

The tumor microenvironment-responsive MNF was prepared by ultrasound-assisted self-assembly. In a typical procedure, the PCL-*b*-pSMA (210 mg) and Fe_3_O_4_ (90 mg) were dissolved in THF (10 mL) completely by oscillation. The mixed solution was then slowly added into an excess alkaline solution (pH ≈ 9, 50 mL) under sonication, followed by dialyzing against the same alkaline solution for 48 h. The dialysis solution was purified by centrifugation (2000 RPM, 10 min), and the supernatant was collected. Finally, the tumor microenvironment-responsive MNF was purified from the supernatant by high-speed centrifugation (100,000× *g*, 20 min). The obtained sediment was collected by lyophilization and stored at 4 °C.

### 3.5. Physicochemical Properties of the Tumor Microenvironment-responsive MNF

First, the morphologies of the Fe_3_O_4_ nanoparticles and MNF were characterized by TEM (Hitachi, HT-7800, Tokyo, Japan) directly, in which the MNF was dispersed into an alkaline solution (pH ≈ 9). Their particle sizes were measured by DLS (Malvern, Nano ZS90, Worcestershire, UK) under an ambient condition. In order to confirm the pH responsiveness of the MNF, we dissolved the MNF into an alkaline solution (pH ≈ 9) and downregulated the surrounding pH value to physiological (pH ≈ 7.43), neutral (pH ≈ 7.02) and weak acidic (pH ≈ 6.67) conditions. Then, we studied their colloidal stability by qualitative observation (digital imaging and TEM) and quantitative analysis (DLS). In addition, the zeta potentials of the MNF in the corresponding buffers (pH ≈ 7.43, pH ≈ 7.02 and pH ≈ 6.67) were characterized simultaneously by DLS. The long colloidal stability of the MNF was also assessed by DLS from 1 to 13 d. The content of the Fe_3_O_4_ in the tumor microenvironment-responsive MNF was measured by TGA (NETZSCH STA 449 F3, Weimar, Germany).

### 3.6. Enhancements of the MNF for MH, FR and MRI under a Neutral Condition

In order to determine the magnetocaloric effect of the MNF, the concentration of Fe ([Fe]) was fixed at 100 μg mL^−1^, which was identified by an inductively coupled plasma mass spectrometer (ICP-MS, Thermo scientific, Xseries II, Waltham, USA). Then, the heating curves of the MNF under physiological (pH ≈ 7.45), neutral (pH ≈ 6.97) and weak acidic (pH ≈ 6.64) conditions were plotted. In this study, an AMF generator (SPG-20AB, ShuangPing Tech. Ltd., Shenzhen, China) with the corresponding frequency (*f*, 114 kHz) and strength (*H*_applied_, 89.9 kA m^−1^) was employed. The inner diameter of the heating coil was 28 mm. The increasing temperature was recorded by a computer-attached fiber optic temperature sensor (FISO, FOT-M, Québec, Canada). Finally, the SAR was calculated by the formula described in a relative study [23].

In addition to the MH, the catalytic potential of MNF under physiological (pH ≈ 7.45), neutral (pH ≈ 6.97) and weak acidic (pH ≈ 6.64) conditions was also studied by detecting the generation of •OH. As the •OH can induce the degradation of MB [47], the study was divided into five groups: MB, MB + H_2_O_2_, MB + H_2_O_2_ + MNF (pH = 7.45), MB + H_2_O_2_ + MNF (pH = 6.97) and MB + H_2_O_2_ + MNF (pH ≈ 6.64). In this study, the concentrations of MB, H_2_O_2_ and MNF were 50 μg mL^−1^, 1 mM and 250 μg mL^−1^. Then, the degradation of MB was observed using digital imaging and monitored using a UV-Vis spectrophotometer (Shimadzu, UV2600, Tokyo, Japan) 2 h later.

The MRI studies were performed with a 3.0-T clinic MRI imaging system (Siemens Trio 3T MRI Scanner, Erlangen, Germany), which was equipped by a micro coil for the transmission and reception of the signal. After dissolving the MNF in an alkaline solution (pH ≈ 8.45), the concentrations of Fe ([Fe]) were identified by ICP-MS first. Then, the study was divided into two groups: physiological condition (pH ≈ 7.42) and neutral condition (pH ≈ 6.99). All groups possessed identifiable [Fe] from 1360 to 42.5 μM.

For the *T*_2_-weighted images, two groups with a series of [Fe] gradients were scanned under these conditions, listed as the following: TR = 5000 ms, TE = 10–90 ms, slice thickness = 3 mm and flip angle = 150°.

### 3.7. Cellular Studies on MH and MH-Induced ROS Generation

In this study, we used a mouse-derived breast cancer cell line, 4T1, purchased from the Chinese Academy of Sciences (Shanghai, China). The 4T1 was cultured using RPMI-1640 medium (Hyclone) containing 10% fetal bovine serum (FBS, every green, Hangzhou, China) and then placed in an incubator at 37 °C with 5% CO_2_ and humidified conditions. The cytotoxicity in vitro was performed by the standard MTT assay. In the research, the biocompatibilities of the PCL-*b*-pSMA and MNF with varied concentrations from 0.25–1 mg mL^−1^ under physiological conditions for 48 h were studied.

To evaluate the efficiency of the MH in vitro, the MNF was dissolved in RPMI-1640 medium (containing 10% FBS and 1% penicillin–streptomycin) at a concentration of 0.2 mg mL^−1^. Considering the effect of *H*_applied_ on the MH, the study was divided into three group, MH-1, MH-2 and MH-3, which were operated under the corresponding *H*_applied_ as 21.2, 31.8 and 42.4 kA m^−1^, respectively. The 4T1 was incubated in a culture dish (35 mm) at a density of 4 × 10^5^ cells per dish in 2 mL of corresponding medium. After incubating for 24 h, the culture medium was replaced by the corresponding medium containing MNF (0.2 mg mL^−1^). Then, the 4T1 was placed in a heating coil with an inner diameter of approximately 38 mm. The applied AMF possessed a constant frequency (*f* = 114 kHz) and varied *H*_applied_ (21.2, 31.8 and 42.4 kA m^−1^). The exposure time under AMF was fixed at 10 min. After MH, the cells were cultured for a prolonged 12 or 24 h. Finally, the cell viabilities were quantified by MTT. After the cell experiment, the heating curves of the MNF (dissolved in RPMI-1640 medium) were recorded in a culture dish (35 mm) under AMF with the same conditions as described for the MH study.

For detecting the ROS generation, DCFH-DA was used following the instructions of the ROS Assay Kit (Beyotime Biotech. Co., Ltd., Shanghai, China). Prior to treatment of the DCFH-DA, 4T1 was incubated in a culture dish (35 mm) at a density of 2 × 10^5^ cells per dish in 2 mL of corresponding medium. Then, the 4T1 was treated by the MNF (0.2 mg mL^−1^) alone for different culture times (12, 24 and 48 h) or MH for 12 h under the established conditions, as described in the MH study. After pretreatment, the culture medium of the 4T1 was replaced by serum-free medium containing DCFH-DA (10 μM) for 40 min. Finally, 4T1 was collected and analyzed by flow cytometry.

## 4. Conclusions

We successfully prepared tumor microenvironment-responsive MNFs by the self-assembly of pH-responsive PCL-*b*-pSMA and superparamagnetic hydrophobic Fe_3_O_4_ nanoparticles for targeted tumor theranostics. As the PCL-*b*-pSMA possessed a p*K*_a_ within a pH range of the tumor microenvironment, the MNF exhibited a great potential for tumor targeting, by maintaining a high colloidal stability in a physiological environment for a long time, forming large, agglomerated clusters in the tumor microenvironment rapidly. Because the tumor microenvironment-responsive MNF showed larger agglomerated clusters in a tumor microenvironment, the close-packed multiple SPIOs with a larger diameter exhibited better performance in reducing the *T*_2_ and increasing SAR simultaneously. Meanwhile, due to the agglomeration of the MNF in the tumor tissue, the encapsulated Fe_3_O_4_ nanoparticles could be exposed from the inner core of the micellar MNF, which induced efficient ER by the interaction between Fe^2+^ and endogenous H_2_O_2_. The further cell experiments confirmed that the tumor microenvironment-responsive MNF displayed excellent biocompatibility, corresponding to its high stability in a physiological environment. However, after applying AMF, the tumor microenvironment-responsive MNF exhibited high efficiency in inhibiting cell proliferation, because of the MH-induced CDT. According to these results, the tumor microenvironment-responsive MNF not only showed high safety in medical areas but also showed itself as a powerful tool to enhance tumor contrast by MRI and improve tumor treatment by combing MH and CDT specifically.

## Figures and Tables

**Figure 1 pharmaceuticals-16-00166-f001:**
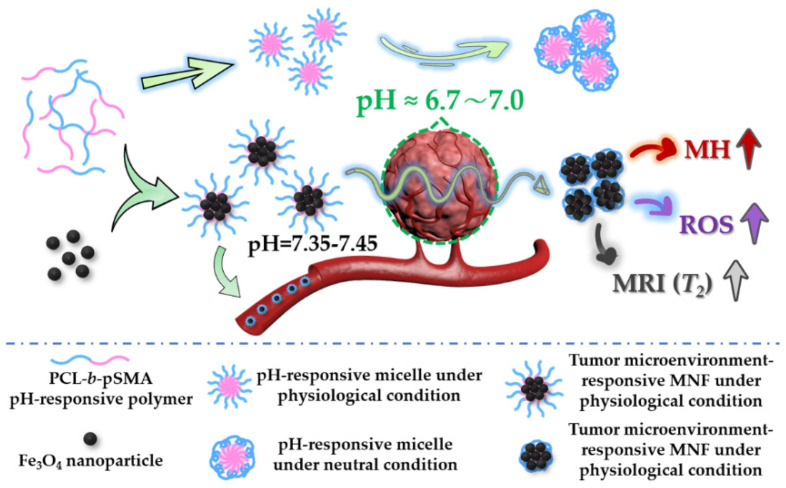
Schematic illustration of the tumor microenvironment-responsive MNF, which was prepared by self-assembly of Fe_3_O_4_ nanoparticles and pH-responsive polymer. According to the phase transition of the pH-responsive micelles under a neutral condition, the tumor microenvironment-responsive MNF could maintain an individual state and form aggregations in tumor microenvironment to improve tumor MH, ROS generation and MRI simultaneously.

**Figure 2 pharmaceuticals-16-00166-f002:**
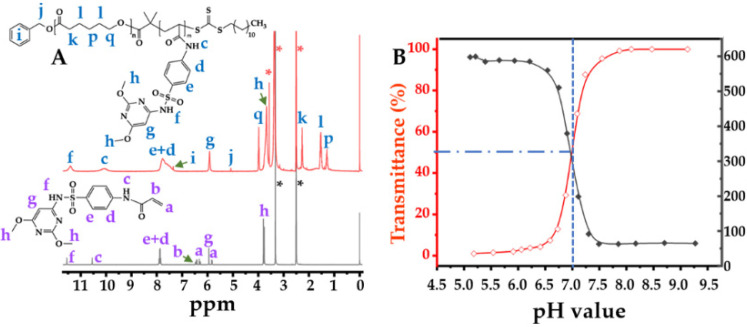
Structure and pH responsive of the PCL-*b*-pSMA: (**A**) ^1^H NMR spectrum of PCL-*b*-pSMA, marking all characteristic peaks; (**B**) pH-dependent transmittance curve and diameter variation of PCL-*b*-pSMA, showing a p*K*_a_ at 7.0.

**Figure 3 pharmaceuticals-16-00166-f003:**
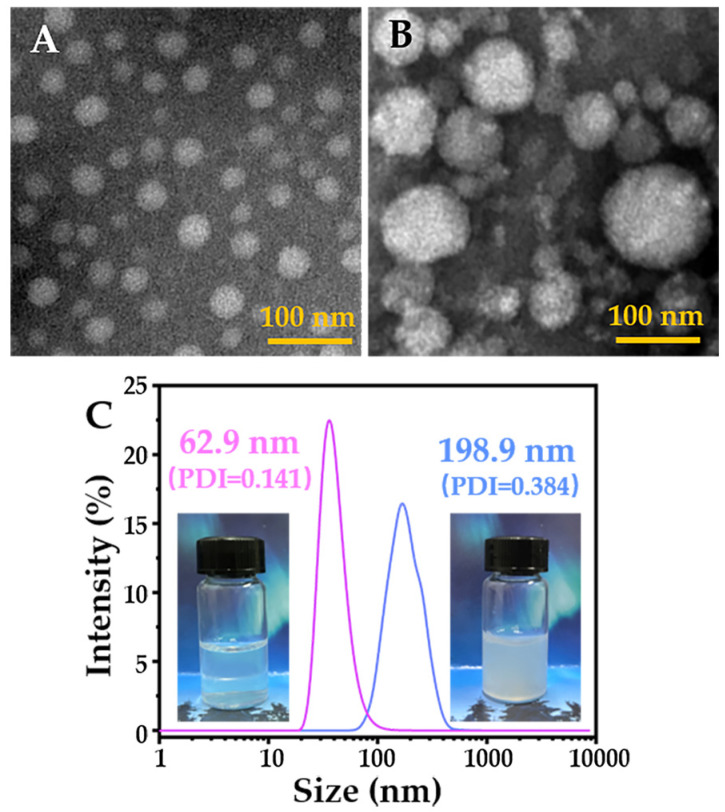
Morphologies (TEM) and particle size distributions (DLS) of PCL-*b*-pSMA micelles in the aqueous phase (0.3 mg mL^−1^) with different pH values: (**A**) TEM result of PCL-*b*-pSMA micelles in the buffer with a physiological pH value (pH = 7.43); (**B**) TEM result of PCL-*b*-pSMA micelles in neutral buffer (pH = 7.05); (**C**) DLS results of the samples in (**A**,**B**).

**Figure 4 pharmaceuticals-16-00166-f004:**
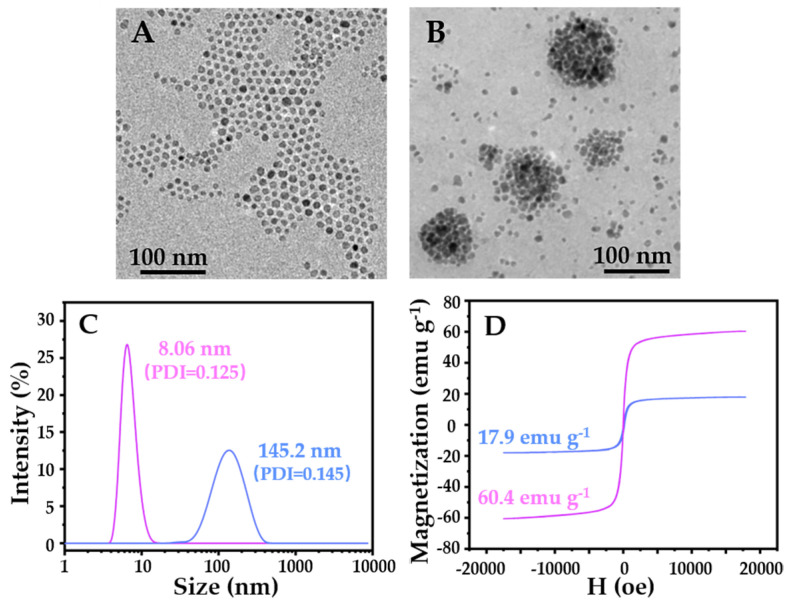
Characterization of the Fe_3_O_4_ nanoparticles and MNF, including the morphologies (TEM), particle size distributions (DLS) and magnetic properties (magnetic hysteresis loops): (**A**) TEM result of the hydrophobic Fe_3_O_4_ nanoparticles in hexane (0.1 mg mL^−1^); (**B**) TEM result of the MNFs in alkaline buffer (pH ≈ 9.0, 0.5 mg mL^−1^); (**C**) DLS results of the samples in (**A**,**B**); (**D**) magnetic properties of the desiccative Fe_3_O_4_ nanoparticles (60.4 emu g^−1^) and MNF (17.9 emu g^−1^).

**Figure 5 pharmaceuticals-16-00166-f005:**
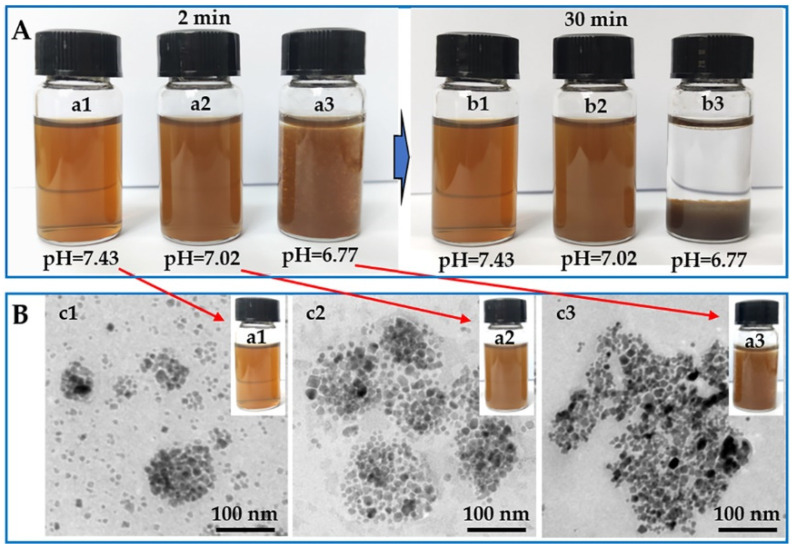
The macroscopic states (digital photo) and microscopic morphologies (TEM) of the tumor microenvironment-responsive MNF in different buffers (0.5 mg mL^−1^): (**A**) digital photos of the MNF for 2 and 30 min; (**B**) TEM results of the MNF for 2 min.

**Figure 6 pharmaceuticals-16-00166-f006:**
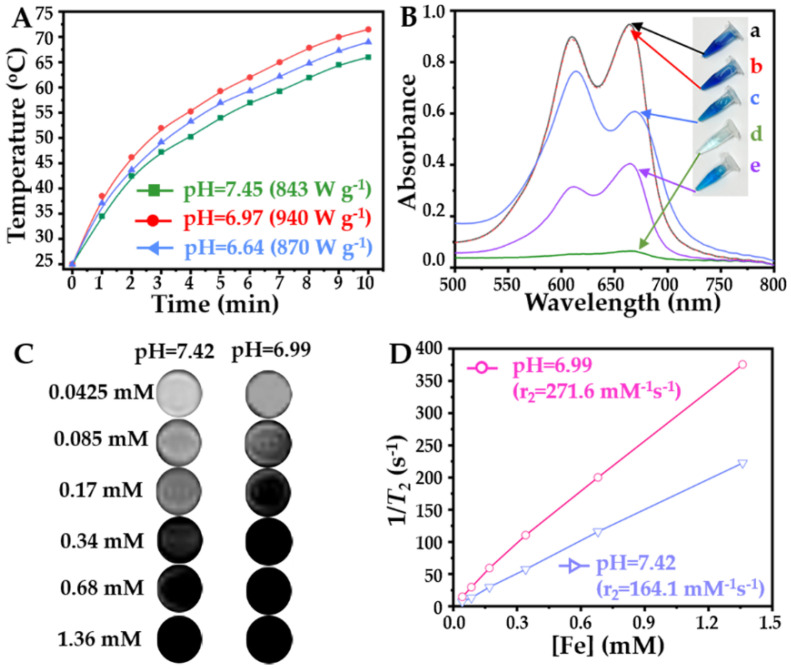
The effect of the tumor microenvironmental pH value on the tumor microenvironment-responsive MNF’s applications, including (**A**) enhanced MH with a concentration of Fe at 100 μg mL^−1^; (**B**) increased ROS generation with the concentration of the MNF at 250 μg mL^−1^; (**C**,**D**) improved *T*_2_ relaxivity of the MRI with varied Fe concentrations from 1.36 to 0.0425 mM.

**Figure 7 pharmaceuticals-16-00166-f007:**
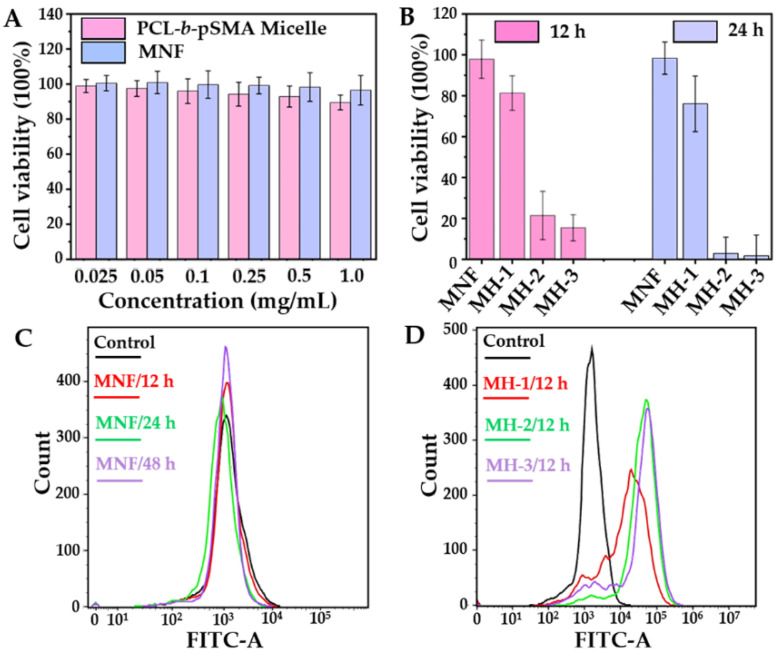
Cellular experiments on the biological effect of the MNF and MH, including the (**A**) biocompatibility of the PCL-*b*-pSMA micelles and MNF; (**B**) MH-induced cell death; intercellular ROS generation by (**C**) MNF (**D**) and MH.

## Data Availability

Data is contained within the article and Appendix A.

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
