# Peer review of "Tumor Microenvironment-Responsive Magnetic Nanofluid for Enhanced Tumor MRI and Tumor multi-treatments"

_pharmaceuticals, 2023, doi:10.3390/ph16020166_

Round 1
This review report has been removed from the review record as it did not conform with MDPI’s standards (https://www.mdpi.com/reviewers#_bookmark11).
Reviewer 2 Report
This is a wonderful article describing a comprehensive examination of the invitro strategies associated with MNF for theragnostic activities in the tumor microenvironment. The authors described the article comprehensively. I would like to give some suggestions to the authors to clarify and describe in their manuscript.
Abstract: Line 21, Put invitro before cell experiments.
2) Put scale bars in Fig.5b (b1 and b2).
3) Since, PCL-B-pSMA are pH-sensitive polymers. How long PCL-b-pSMA micelles or MNFs become stable at ph 7.4? Did the authors perform stability tests as a function of time at pH 7.4? It would be great to figure out the stability at pH 7.4 for a few hours (5h or 6h) and see the particle stability in terms of size distribution.
Author Response
Response to the second referee's comments
Comment 1: “Abstract: Line 21, Put invitro before cell experiments.”
Answer: Thank you very much for your attention on our research. According to your comment, I have edited the abstract to adjust the in vitro experiments before the cell experiments. The relative details have been remarked by red color in the “version of revised manuscript for reviewers”. I hope that reviewer would like the new version of our manuscript.
Comment 2: “Put scale bars in Fig.5b (b1 and b2).”
Answer: Thank you very much to correct our negligence. Based on the comment, we modified the Figure 5b, adding scale bars as "100 nm". I am really sorry for our negligence. I hope that reviewer would like the new version of our manuscript.
Comment 3: “Since, PCL-B-pSMA are pH-sensitive polymers. How long PCL-b-pSMA micelles or MNFs become stable at ph 7.4? Did the authors perform stability tests as a function of time at pH 7.4? It would be great to figure out the stability at pH 7.4 for a few hours (5h or 6h) and see the particle stability in terms of size distribution.”
Answer: Thank you very much for your attention on our research. Based on the comment, I supplemented the Figure S5 to Support Information, and added ralative description at the part of "2.2 Characterization of tumor stroma responsive MNF", which have been remarked by red color in the “version of revised manuscript for reviewers”. I hope that reviewer would like the new version of our manuscript. Thank you very much.

Reviewer 3 Report
This work presents interesting results. There are just some issues as follows that should be addressed to consider the paper for the publication:
- please specify in the procedure how many mg of SPIONs were used in the nanofluid. The details of the MNFs are not precise. In line 200, the authors deliver information about the Fe3O4 use, so please specify the concentration of magnetic nanocarriers in the 1 mL volume.
- Bare Fe3O4 are hydrophilic, so please specify the functional groups responsible for the hydrophobic effect or present conditions in the text/table/schematic diagram.
- Fig. 4a shows the TEM image of Fe3O4 and specifies 8 nm in diameter in the text. However, the image clearly shows nanoparticles having a different diameters. Please show the histogram and add the mean size.
- The list of chemicals shows only the Iron (III) salt. Therefore it is not clear how the authors obtained the Fe3O4 which is an oxide containing (II) and (III) valent iron. Please deliver details of the synthesis way and the magnetization of particles and the confirmation of the chemical composition of the SPIONs. In the text, the authors describe the SPIONs, so the hysteresis curves confirming the superparamagnetism should be presented.
- SAR values should be presented in W/g, not g/W. Please check the obtained values and correct units. What is the concentration of SPIONs in the sample tested by MH. Please describe more precisely the effect of pH on SAR values - interactions between groups etc.
- Please specify if the proposed conditions of MH are within the Atkinson-Brezovich safe limit.
- In general, specify in numbers the conditions (concentration etc., when presenting graphs), so the procedure will be more transparent.
- Zeta potential studies showing the long-term stability of the samples would be recommended to be shown here.
Author Response
Response to the third referee's comments
Comment 1: “please specify in the procedure how many mg of SPIONs were used in the nanofluid. The details of the MNFs are not precise. In line 200, the authors deliver information about the Fe3O4 use, so please specify the concentration of magnetic nanocarriers in the 1 mL volume.”
Answer: Thank you very much for your attention on our research. According to your comment, I added a description of the specific amount of SPIONs at the part of "2.2 Characterization of tumor stroma responsive MNF". Similarly, I burnished the preparation process of MNFs at the part of "3.4 Preparation of tumor stroma responsive MNF". In the paragraph, I added the specific dosage of various materials, including PCL-b-pSMA and Fe3O4. The relative description has been remarked by red color in the “version of revised manuscript for reviewers”. I hope that reviewer would like the new version of our manuscript. Thank you very much again.
Comment 2: “Bare Fe3O4 are hydrophilic, so please specify the functional groups responsible for the hydrophobic effect or present conditions in the text/table/schematic diagram.”
Answer: Thank you very much for your comment. As I know, SPION can be synthesized by several methods, including sol-gel synthesis, co-precipitation, hydrothermal synthesis and thermal decomposition. Among these methods, the thermal decomposition method, reported by Sun Shouheng, has been demonstrated as the most effective approach for preparing high-quality magnetic nanocrystals. According to Sun’s studies [1], the monodispersed Fe3O4 nanoparticles could be synthesized by reaction of Fe(acac)3, oleic acid, oleylamine and a long-chain alcohol in organic-phase (phenyl ether or benzyl ether). Furthermore, the reaction has been extended by Sun to the synthesis of MFe2O4 nanoparticles (M = Co, Ni, Mn, etc) with tunable sizes from 3 to 20 nm in diameter by adding a different metal acetylacetonate precursor to the mixture of Fe(acac)3, 1,2-hexadecanediol, oleic acid and oleylamine in benzyl ether [2]. According to the method, high-boiling-point nonpolar solvents (phenyl ether or benzyl ether) were used as reaction media, which could effectively separate the nucleation and subsequent growth process, consequently suppressing the particle size distribution. At the same time, water was excluded from the reaction system, leading to a clearly defined composition of the iron oxide nanoparticles, as water could form complicated compounds with ferric ions. However, the direct products of thermal decomposition approach are soluble in organic solvents, as we described in our manuscript. Therefore, the monodispersed Fe3O4 nanoparticles involved in our manuscript are hydrophobic nanoparticles. For their application in biomedical areas (MH and MRI), the post-preparative procedures were necessary, which were described by many relative researches [3-6]. However, we still appreciate your attention on our study. Thanks for your comment again.
Comment 3: “Fig. 4a shows the TEM image of Fe3O4 and specifies 8 nm in diameter in the text. However, the image clearly shows nanoparticles having a different diameters. Please show the histogram and add the mean size.”
Answer: Thank you very much for your attention on our research. In fact, we described mean size of Fe3O4 in our original manuscript, which was shown in Figure 4C. We did not analyze mean size and size distribution (PDI) of Fe3O4 nanoparticles directly by measuring nanoparticle diameters in Figure 4A, because DLS was a powerful tool to study mean size and size distribution (PDI) of magnetic nanoparticles [4,6]. By DLS, the mean size and size distribution (PDI) of Fe3O4 nanoparticles can be observed easily. Therefore, we presented mean size of Fe3O4 nanoparticles by histogram, as shown in Figure 4C, which was 8.06 nm.
Comment 4: “The list of chemicals shows only the Iron (III) salt. Therefore it is not clear how the authors obtained the Fe3O4 which is an oxide containing (II) and (III) valent iron. Please deliver details of the synthesis way and the magnetization of particles and the confirmation of the chemical composition of the SPIONs. In the text, the authors describe the SPIONs, so the hysteresis curves confirming the superparamagnetism should be presented.”
Answer: Thanks so much for your attention on our research. Based on Sun's method, only Iron(III) acetylacetonate [Fe(acac)3] is used for synthesis of Fe3O4 nanoparticles. According to your valuable advice, the preparation process is supplemented in the revised version, as described at the part of "3.1 Materials ".
Unfortunately, we can’t characterize the structure and chemical composition of Fe3O4 nanoparticles in the next month. Because of COVID-19, our laboratory has been closed in advance. I am very sorry for this situation. However, the method, thermal decomposition, for preparing Fe3O4 nanoparticles has been adopted widely in many studies [3-9]. Therefore, the Fe3O4 nanoparticles, described in the manuscript, should be superparamagnetic nanoparticles with the cubic spinel structure.
Although we can’t provide data on structure and chemical composition of Fe3O4 nanoparticles, magnetic hysteresis loops of Fe3O4 nanoparticles and MNFs can be presented in the revised manuscript, as shown in Figure 4D. In fact, magnetic properties of Fe3O4 nanoparticles and MNFs were characterized during the whole research. As it does not affect the conclusion of the paper, we omitted the results in the original version. In the revised manuscript, the results were supplemented in Figure 4 (Figure 4D). Moreover, we also added relevant descriptions in the “2.2 Characterization of tumor stroma responsive MNF" and "3.5 Physicochemical properties of tumor stroma responsive MNF" sections of the article. All relative details have been remarked by red color in the “version of revised manuscript for reviewers”. I hope that reviewer would like the new version of our manuscript.
Comment 5: “SAR values should be presented in W/g, not g/W. Please check the obtained values and correct units. What is the concentration of SPIONs in the sample tested by MH. Please describe more precisely the effect of pH on SAR values - interactions between groups etc.”
Answer: Thank you very much to review our paper so carefully and point out a mistake. In the revised manuscript, the mistake has bee corrected.
And in the MH experiment, we used inductively coupled plasma mass spectrometry (ICP-MS) to measure the concentration of Fe, which was 100 μg mL-1. According to your comments, we have added relative description in the "2.3 Advantages of tumor stroma responsive MNF on tumor treatment and diagnosis" to emphasize the concentration of Fe in this experiment. In addition, we further elaborated on the influence of pH on SAR values in this section. All relevant content has been marked by red color in the “version of revised manuscript for reviewers”. I hope that reviewer would like the new version of our manuscript. Thank you very much.
Comment 6: “Please specify if the proposed conditions of MH are within the Atkinson-Brezovich safe limit.”
Answer: Thank you very much for your attention on our research. Based on your suggestion, we have reviewed the relevant literature on the Atkinson-Brezovich limit [10-13]. As far as we know, the Atkinson-Brezovich safe limit was developed to ensure patient safety. The highest acceptable value was established to be H × f = 4.85 × 108 Am−1s−1. H × f involves the following expression for the rate of heat production per unit of tissue volume for a cylindrical body:
where ? is the electrical conductivity of the tissue, ?0 the vacuum permeability, the radio (?) of the cylinder and H and f the magnetic field intensity and frequency, respectively.
In addition, on the premise of calculating the magnetic fluid parameters, the power dissipation equation [12,13] is also related to the Atkinson-Brezovich safe limit:
where P is the heat dissipation value, μ0 the magnetic field constant, χ′′ the magnetic susceptibility (imaginary part), f the frequency of the applied magnetic field, and H the strength of the applied magnetic field.
In accordance with our previously experience, magnetic hyperthermia treatments were performed in the magnetic field shown in Table 1 in the described magnetic hyperthermia configuration.
Table 1 Applied electromagnetic fields and H×f values according to the experimental groups
Group Name |
f (kHz) |
H (kAm−1) |
H × f (103 Am−1s−1) |
|
|
MH-1 |
114 |
21.2 |
2.4168 |
||
MH-2 |
114 |
31.8 |
3.6252 |
||
MH-3 |
114 |
42.4 |
4.8336 |
Apparently, all values of H × f obtained from our experiments are much lower than the Atkinson-Brezovich safe limit (4.85 × 108 Am−1s−1). Thus, the MH conditions for our experiments are within Atkinson-Brezovich safe limit. Thanks for your advice again.
Comment 7: “In general, specify in numbers the conditions (concentration etc., when presenting graphs), so the procedure will be more transparent.”
Answer: Thanks for your suggestion, we added all the details about the concentration in the relevant figures and description, and it has been marked in red. I hope that reviewer would like the new version of our manuscript.
Comment 8: “Zeta potential studies showing the long-term stability of the samples would be recommended to be shown here.”
Answer: Your advice is really insightful. In the revised manuscript, we supplemented the zeta potential results, which were characterized during the whole research. Because it did not affect the conclusion of the paper, we omitted the results in the original version. In the revised manuscript, the results were listed in Figure S7. We also supplemented relative description in the “2.2 Characterization of tumor stroma responsive MNF" and "3.5 Physicochemical properties of tumor stroma responsive MNF". Thank you very much again.
Reference:
- Sun, S.; Zeng, H. Size-Controlled Synthesis of Magnetite Nanoparticles. J. Am. Chem. Soc. 2002, 124, 8204-8205.
- Sun, S.; Zeng, H.; Robinson, D.B.et al. Monodisperse MFe2O4 (M = Fe, Co, Mn) Nanoparticles. J. Am. Chem. Soc. 2004, 126, 273-279.
- Hu, F.Q.; Wei, L.; Zhou, Z. et al. Preparation of Biocompatible Magnetite Nanocrystals for In Vivo Magnetic Resonance Detection of Cancer. Adv. Mater. 2006, 18, 2553–2556.
- Lu, J.; Ma, S.; Sun, J.; et al. Manganese ferrite nanoparticle micellar nanocomposites as MRI
contrast agent for liver imaging. Biomaterials 2009, 30, 2919–2928.
- Fe3O4 nanoparticles-loaded PEG-PLA polymeric vesicles as labels for ultrasensitive immunosensors. Biomaterials 2010, 31, 7332-7339.
- Qu, Y.; Li, J.; Ren, J.; et al. Enhanced magnetic fluid hyperthermia by micellar magnetic nanoclusters composed of MnxZn1–x Fe2O4 nanoparticles for induced tumor cell apoptosis. ACS Appl. Mater. Interfaces 2014, 6, 16867-16879.
- Ai, H.; Flask, C.; Weinberg, B.; et al. Magnetite-loaded polymeric micelles as ultrasensitive magnetic-resonance probes. Adv. Mater. 2005, 17, 1949–1952.
- Yang, X.; Grailer, J.J.; Rowland, I.J.; et al. Multifunctional SPIO/DOX-loaded wormlike polymer vesicles for cancer therapy and MR imaging. Biomaterials 2010, 31, 9065-9073.
- Lim, E.K.; Yang, J.; Dinney, C.P.N.; et al. Self-assembled fluorescent magnetic nanoprobes for multimode-biomedical. Biomaterials 2010, 31, 9310-9319.
- Atkinson, W.J.; Brezovich, I.A.; Chakraborty, D.P. Usable Frequencies in Hyperthermia with Thermal Seeds. IEEE. Trans. Biomed. Eng. 1984, BME-31, 70-75.
- Herrero de la Parte, B.; Rodrigo, I.; Gutiérrez-Basoa, J.; et al. Proposal of New Safety Limits for In Vivo Experiments of Magnetic Hyperthermia Antitumor Therapy. Cancers 2022, 14, 3084.
- Laurent, S.; Dutz, S.; Häfeli, U.O.; et al. Magnetic fluid hyperthermia: Focus on superparamagnetic iron oxide nanoparticles. Adv. Colloid Interface Sci. 2011, 166, 8-23.
- Rosensweig, R.E. Heating magnetic fluid with alternating magnetic field. J. Magn. Magn. Mater. 2002, 252, 370-374.
Round 2
This review report has been removed from the review record as it did not conform with MDPI’s standards (https://www.mdpi.com/reviewers#_bookmark11).